# High-Layer Attention Pruning with Rescaling

**Songtao Liu*   Peng Liu**
*The Pennsylvania State University*

**Reviewed on OpenReview:** *https://openreview.net/forum?id=jkPBIxYmWE*

## Abstract

Pruning is a highly effective approach for compressing large language models (LLMs), significantly reducing inference latency. However, conventional training-free structured pruning methods often employ a heuristic metric that indiscriminately removes some attention heads across all pruning layers, without considering their positions within the network architecture. In this work, we propose a novel pruning algorithm that strategically prunes attention heads in the model's higher layers. Since the removal of attention heads can alter the magnitude of token representations, we introduce an adaptive rescaling parameter that calibrates the representation scale post-pruning to counteract this effect. We conduct comprehensive experiments on a wide range of LLMs, including LLaMA3.1-8B, Mistral-7B-v0.3, Qwen2-7B, and Gemma2-9B. Our evaluation includes both generation and discriminative tasks across 27 datasets. The results consistently demonstrate that our method outperforms existing structured pruning methods. This improvement is particularly notable in generation tasks, where our approach significantly outperforms existing baselines. Code is available at `https://github.com/SongtaoLiu0823/HARP`.

## 1 Introduction

Large language models (LLMs) (Touvron et al., 2023; OpenAI et al., 2023; Jiang et al., 2023; Yang et al., 2024; Gemma2-Team et al., 2024), pre-trained on extensive text data, have achieved surprising performance in downstream tasks such as information retrieval (Asai et al., 2024), code generation (Guo et al., 2024), and mathematical reasoning (Wang et al., 2023; Yang et al., 2023b; Huang et al., 2024). These LLMs, however, contain a huge number of parameters, resulting in substantially slower inference speed compared to their smaller counterparts.

Many efforts have leveraged pruning algorithms (Frantar & Alistarh, 2023; Jaiswal et al., 2023; Xia et al., 2024; Ashkboos et al., 2024; Xu et al., 2024; Jaiswal et al., 2024a; Zhang et al., 2024b; Dong et al., 2024b; Yin et al., 2024a;b; Zhao et al., 2024) to reduce redundant parameters and lower inference latency without significantly compromising the performance. Among these, training-free structured pruning methods (Ma et al., 2023; An et al., 2024; Sun et al., 2024b) estimate the importance of parameters using predefined pruning metrics, allowing them to remove less important components without retraining. While most pruned parameters typically come from feedforward networks, these methods also remove some low-importance attention heads. This is especially beneficial for long-context tasks, where the attention mechanism becomes the main computational bottleneck due to its quadratic time complexity with respect to sequence length.

However, current pruning algorithms generally prune attention heads across all targeted layers, either by uniformly removing the same heads from each layer (Ma et al., 2023) or by adaptively pruning more heads in certain layers and fewer in others (An et al., 2024). These approaches do not consider which attention head positions are actually more redundant and thus better suited for pruning. We argue that the pruning metrics used in these methods are not accurate enough to reliably identify the most redundant attention heads. In this work, we aim to address the following question:

> *Can we develop a structured pruning algorithm that effectively removes attention heads?*

---

*Correspondence to: Songtao Liu <skl5761@psu.edu>.

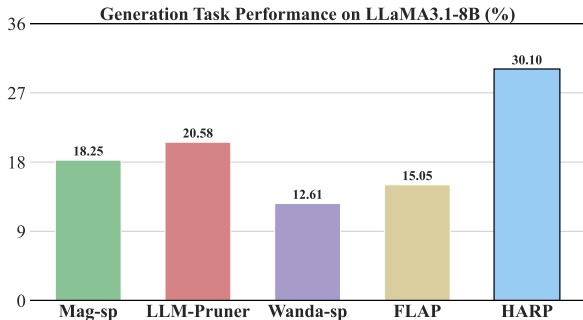 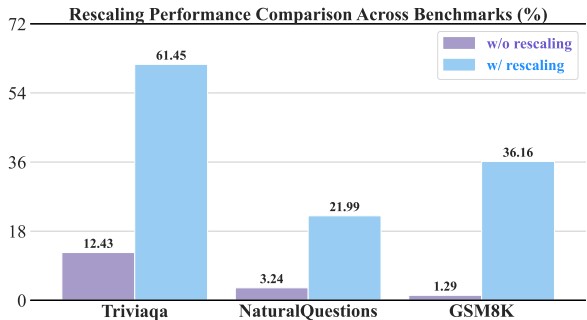

Figure 1: **Left**: Performance comparison of our method against state-of-the-art baselines on LLaMA3.1-8B, demonstrating superior generation capabilities. **Right**: Ablation study on the rescaling parameter across benchmarks, showing that applying rescaling consistently leads to significant performance improvements over the variant without rescaling.

We approach this problem by analyzing the relative importance of different layers in LLMs. Recent studies (Men et al., 2024; Zhong et al., 2024; Gromov et al., 2024; He et al., 2024; Siddiqui et al., 2024) demonstrate that pruning blocks or entire layers in the high layers of LLMs often yields better performance than pruning those in the low layers. Building on these findings, we propose a training-free structured pruning algorithm, called **HARP** (**H**igh-layer **A**ttention **R**escaled **P**runing), that removes the attention heads, specifically the query and key parameters, in the higher layers. After pruning, the attention mechanism in these layers can be skipped entirely. However, bypassing the attention computation may alter the output magnitude, as the value representations are no longer combined through weighted averaging based on attention scores over the full sequence. To mitigate this effect, we introduce a rescaling parameter in the attention residual block to calibrate the token representation magnitude after pruning.

We evaluate the effectiveness of our approach by applying it to LLaMA3.1-8B, Mistral-7B-v0.3, Qwen2-7B, and Gemma2-9B. Extensive experiments on 27 representative downstream tasks show that our method outperforms popular training-free structured pruning algorithms, including Magnitude Pruning (Han et al., 2015), LLM-Pruner (Ma et al., 2023), FLAP (An et al., 2024), and Wanda (Sun et al., 2024b). As illustrated on the left side of Figure 1, our method, HARP, significantly outperforms these baselines on generation tasks under the same attention head pruning ratio. More notably, the results on the right side of Figure 1 show that applying rescaling significantly improves performance compared to the unscaled variant, demonstrating the importance of rescaling for properly adjusting the representation magnitude.

## 2 High-Layer Attention Pruning

In this section, we first demonstrate the importance of pruning attention heads in LLMs as part of structured pruning algorithms to reduce latency in downstream long-context tasks (Section 2.1). Next, we provide a theoretical and empirical explanation for why attention heads in higher layers are less important (Section 2.2). Finally, we present the details of our pruning algorithm (Section 2.3).

### 2.1 Pruning Attention Heads for Long-Context Modeling

Most existing structured pruning algorithms for LLMs primarily focus on pruning parameters in the Feed-Forward Network (FFN), as they contain the majority of the model's parameters. By pruning the FFN, we can significantly reduce the latency. However, pruning attention heads is also crucial. In long-context tasks such as retrieval-augmented generation (RAG), long-document summarization, and many-shot in-context learning (ICL), the attention mechanism often becomes the primary computational bottleneck due to its complexity $O\left(N^2\right)$ with respect to the sequence length $N$. We provide a detailed analysis of the time complexity associated with each component of LLMs.

**Theoretical Computation Complexity Analysis.** Let $N$ be the sequence length, $d$ the hidden size, and $d_{\text{ff}}$ the intermediate dimension in the FFN. In a standard transformer block, the linear projections in the attention module, including query, key, value, and output projections, have a time complexity of $O\left(Nd^2\right)$. The self-attention mechanism, which includes computing attention scores and aggregating values, has a time complexity of $O\left(N^2d\right)$. The FFN, consisting of two linear transformations with an intermediate dimension $d_{\text{ff}}$, has a time complexity of $O\left(Ndd_{\text{ff}}\right)$.

When $N < d_{\text{ff}}$, the FFN's $O\left(Ndd_{\text{ff}}\right)$ complexity becomes the main time bottleneck. Conversely, when $N > d_{\text{ff}}$, the $O\left(N^2d\right)$ complexity of the self-attention mechanism dominates. In long-context tasks, prompt lengths can reach up to 2 million tokens (Jin et al., 2025), making the operations in the self-attention module extremely time-expensive. By pruning attention heads (Bansal et al., 2022), we can achieve significant improvements in computational efficiency for long-context tasks while only pruning a small fraction of the model's parameters.

## 2.2 High-Layer Attention Heads Are Less Important

While many structured pruning algorithms focus on pruning attention heads, they often overlook the attention head position within the network. In this section, we show that attention heads in higher layers are typically less important than those in lower layers.

**Theoretical Analysis.** In this subsection, we formalize what we mean by "tokens becoming similar" in a layer and how this affects the usefulness of self-attention. Let $\mathbf{H}^{(\ell)} \in \mathbb{R}^{N \times d}$ denote the hidden representation matrix at layer $\ell$, whose $i$-th row $\mathbf{h}_i^{(\ell)}$ corresponds to the representation of the $i$-th token. Throughout this paper, when we say that "the token representations are (almost) the same", we specifically mean that their pairwise cosine similarities are close to 1, i.e., all token vectors are nearly parallel in the representation space.

**Definition 1.** *The average pairwise cosine similarity among tokens in a matrix* $\mathbf{H} \in \mathbb{R}^{N \times d}$ *is defined as*

$$\text{Sim}(\mathbf{H}) := \frac{2}{N(N-1)} \sum_{1 \leq i < j \leq N} \frac{\mathbf{h}_i^\top \mathbf{h}_j}{\|\mathbf{h}_i\|_2 \|\mathbf{h}_j\|_2}. \tag{1}$$

A value $\text{Sim}(\mathbf{H}) \approx 1$ indicates that all token representations are almost colinear (up to scaling), while a smaller value means that the representations point in more diverse directions. This quantity therefore measures the global level of similarity among token representations.

**Theorem 1.** *When* $\text{Sim}(\mathbf{H}^{(\ell)}) \to 1$, *token representations in layer* $\ell$ *become nearly parallel. Since queries and keys are linear projections of* $\mathbf{H}^{(\ell)}$, *their dot products across positions become almost constant, so the row-wise softmax yields nearly uniform attention weights. The head thus degenerates into averaging value vectors and cannot distinguish tokens. Conversely, if* $\text{Sim}(\mathbf{H}^{(\ell)})$ *is bounded away from* 1, *attention remains non-uniform and can model meaningful token-to-token dependencies.*

A formal proof of Theorem 1 is provided in Appendix A. Intuitively, as the hidden states within a layer become more and more aligned (large $\text{Sim}(\mathbf{H}^{(\ell)})$), the attention operation behaves like a simple averaging operator that mixes already similar representations. This phenomenon is analogous to over-smoothing in deep GNNs, where repeated averaging causes node embeddings to collapse into a low-dimensional subspace. As a result, attention heads in very high layers, where representations are highly aligned, are less informative than those in lower layers where representations remain more diverse.

**Empirical Analysis.** Recent studies (Men et al., 2024; Zhong et al., 2024; Gromov et al., 2024; He et al., 2024; Siddiqui et al., 2024; Liu et al., 2024b; Zhang et al., 2024a; Jaiswal et al., 2024b; Chen et al., 2024b; Kim et al., 2024a) have introduced various heuristic metrics to evaluate the importance of blocks within each layer of LLMs. A common finding across these works is that the parameters in lower layers tend to be more important than those in higher layers, lending empirical support to our theoretical analysis. One such metric is the Block Importance (BI) score (Men et al., 2024):

$$\text{BI}^{(\ell)} = 1 - \mathbf{E}_{\mathbf{H},t} \frac{\left\langle \mathbf{H}_t^{(\ell)}, \mathbf{H}_t^{(\ell+1)} \right\rangle}{\left\| \mathbf{H}_t^{(\ell)} \right\|_2 \left\| \mathbf{H}_t^{(\ell+1)} \right\|_2}. \tag{2}$$

---

**Algorithm 1** Top-down $\alpha$ search for HARP

---

**Input**: Base LLM, number of layers to prune $P$, search values $\alpha \in \{0, 0.1, \ldots, 1.0\}$
**Initialize**: $\boldsymbol{\alpha}_{\text{best}} \in \mathbb{R}^P = [1.0, 1.0, \ldots, 1.0]$
$L$ = total number of layers in the model
**for** $\ell = L - 1$ down to $L - P$ **do**
    $\text{PPL}_{\text{best}} = \infty$
    **for** each $\alpha \in \{0, 0.1, \ldots, 1.0\}$ **do**
        Set rescaling parameter $\boldsymbol{\alpha}_{\text{best}}[L - 1 - \ell] = \alpha$ for layer $\ell$
        $\text{PPL} = \text{Perplexity}(\text{LLM}, \boldsymbol{\alpha}_{\text{best}})$
        **if** $\text{PPL} < \text{PPL}_{\text{best}}$ **then**
            $\text{PPL}_{\text{best}} = \text{PPL}$
            $\boldsymbol{\alpha}_{\text{best}}[L - 1 - \ell] = \alpha$
        **end if**
    **end for**
**end for**
**Output**: Optimal rescaling parameters $\boldsymbol{\alpha}_{\text{best}} \in \mathbb{R}^P$ for each pruned layer

---

The BI score typically decreases with increasing layer index $\ell$, indicating the reduced impact of higher-layer parameters. In this work, we use these empirical observations as additional evidence that pruning higher-layer attention heads tends to be less harmful than pruning lower-layer ones, and we explicitly incorporate this layer-wise asymmetry into our pruning design.

## 2.3 Methodology

In this section, we introduce **HARP**, a novel structured pruning algorithm specifically designed to remove attention heads in the higher layers of LLMs. Our approach focuses on pruning only the query and key parameters in these layers. As a result, we can bypass the self-attention mechanism for these pruned layers. Concretely, when the attention of a layer is fully pruned, the layer no longer computes attention scores or performs aggregation; instead, it only applies the value and output projections. The computation within the self-attention module can then be formulated as $\mathbf{H} = \mathbf{H}\mathbf{W}_V \mathbf{W}_O$.

**Rescaling.** The aggregation step in self-attention computes, for each token, a weighted average of value vectors over all positions in the sequence. This operation not only mixes information across tokens but can also change the overall scale of the hidden representations (Kipf & Welling, 2017). We measure this scale using the Frobenius norm of the hidden state matrix $\mathbf{H}$.

**Proposition 1.** *Let $\mathbf{H} \in \mathbb{R}^{N \times d}$ denote the token representations, and let $\hat{\mathbf{A}} \in \mathbb{R}^{N \times N}$ be a row-normalized attention matrix (i.e., each row is non-negative and sums to 1). Consider the aggregation step in self-attention defined by $\mathbf{H}' = \hat{\mathbf{A}}\mathbf{H}$. In general, we have*

$$\|\mathbf{H}'\|_F \neq \|\mathbf{H}\|_F, \tag{3}$$

*and equality holds if and only if every row of $\hat{\mathbf{A}}$ is a one-hot vector (so that each output token simply copies one input token without averaging). This means that, except for this degenerate case, the aggregation step changes the magnitude of token representations.*

A detailed proof of Proposition 1 is also given in Appendix A ("Proof of Proposition 1"). The proposition implies that removing the aggregation step (by pruning queries and keys and bypassing attention) will in general alter the scale of the hidden states compared to the original model.

Based on this proposition, we argue that the magnitude of token representations can change after pruning, since the aggregation step is bypassed. Inspired by normalization approaches (Ioffe & Szegedy, 2015; Ba et al., 2016), which maintain the magnitude of token representations within a stable range to prevent gradient explosion or vanishing during pretraining, we similarly adjust the magnitude after pruning. This adjustment keeps the token representation magnitude within a stable range relative to their pre-pruning state, helping

Table 1: Comparison of FLAP with existing pruning algorithms on LLMs.

| Method | Weight Update | Calibration Data | Pruning Metric |
|--------|:---:|:---:|:---:|
| Magnitude | ✗ | ✗ | $\lvert\mathbf{W}_{ij}\rvert$ |
| Wanda | ✗ | ✓ | $\lvert\mathbf{W}_{ij}\rvert \cdot \lVert\mathbf{X}_j\rVert_2$ |
| FLAP | ✗ | ✓ | $\frac{1}{N-1}\sum_{n=1}^{N}\left(\mathbf{X}_{n,j,:}^{\ell} - \overline{\mathbf{X}}_{:,j,:}^{\ell}\right)^2 \cdot \left\lVert\mathbf{W}_{:,j}^{\ell}\right\rVert_2^2$ |
| HARP | ✗ | ✓ | ppl |

preserve model performance during inference. We introduce a rescaling parameter $\alpha$ into the residual block of the self-attention module as $\mathbf{H} = \mathbf{H} + \alpha\mathbf{H}\mathbf{W}_V\mathbf{W}_O$.

Determining the optimal value of $\alpha$ is challenging, as we treat $\alpha$ as a non-differentiable, layer-wise hyperparameter rather than a trainable parameter. Instead of retraining the model, we propose a simple and efficient greedy search strategy that selects $\alpha$ by directly evaluating the perplexity of the pruned LLM on a held-out corpus.

To simplify the search process, we adopt a top-down approach: starting from the topmost layer (layer $L$), we determine the optimal $\alpha$ for that layer while keeping all other layers unpruned or fixed; we then fix this value and proceed to search for $\alpha$ in the next lower layer. This layer-by-layer greedy strategy forms a sequential for-loop that significantly reduces the overall search space, as shown in Algorithm 1.

In practice, we implement this procedure as a grid search over a small set of candidate values $\{0, 0.1, \ldots, 1.0\}$, which requires only a modest number of forward passes (perplexity evaluations) and is therefore much cheaper than any training-based method. Designing more advanced search strategies (e.g., Bayesian optimization or differentiable relaxations) is an interesting open direction for future work.

## 3 Experiments

### 3.1 Experimental Setup

**Baselines.** We evaluate HARP against four baselines: Magnitude pruning (Han et al., 2015), LLM-Pruner (Ma et al., 2023), FLAP (An et al., 2024), and Wanda (Sun et al., 2024b), as shown in Table 1. We build upon the FLAP to extend Magnitude Pruning and Wanda to structured pruning, referring to these variants as Mag-sp and Wanda-sp. Mag-SP, LLM-Pruner, and Wanda-SP prune attention heads uniformly across all targeted layers by removing the same number of heads from each layer, whereas FLAP adaptively prunes more heads in some layers and fewer in others.

**Model Setting.** We conduct experiments on four GQA-based LLMs: LLaMA3.1-8B (Touvron et al., 2023), Mistral-7B-v0.3 (Jiang et al., 2023), Qwen2-7B (Yang et al., 2024), and Gemma2-9B (Gemma2-Team et al., 2024). For LLaMA3.1-8B and Mistral-7B-v0.3, we set the attention head pruning ratio to 1/4, whereas for Qwen2-7B and Gemma2-9B, it is set to 1/7. Therefore, we use HARP to prune 8 layers for LLaMA3.1-8B, 4 layers for Qwen2-7B, 8 layers for Mistral-7B-v0.3, and 6 layers for Gemma2-9B. We find that our method achieves a smaller pruning ratio compared to the baselines. To address this and ensure a fair comparison, we integrate FLAP with our HARP, leveraging FLAP to prune FFN parameters in our implementation. Aside from this, FFN parameters are not pruned in the baseline methods.

**Baseline Implementation.** It is important to note that the FLAP official implementation[1] does not support GQA-based LLMs. Therefore, we use a modified implementation[2] for our experiments. Furthermore, the official implementation[3] of LLM-Pruner is available only for LLaMA3.1-8B. Our HARP is compatible with any attention mechanism.

---

[1] https://github.com/CASIA-IVA-Lab/FLAP
[2] https://github.com/nyunAI/FLAP
[3] https://github.com/horseee/LLM-Pruner

Table 2: Performance on generation tasks across LLaMA3.1-8B, Mistral-7B-v0.3, Qwen2-7B, and Gemma2-9B (%). Best is **bold**.

| Method | Parameter Pruning Ratio | Attention Head Pruning Ratio | TriviaQA | NaturalQuestions | GSM8K | MATH-hard | BBH | Average |
|---|---|---|---|---|---|---|---|---|
| **LLaMA3.1-8B** | | | | | | | | |
| Dense | 0 | 0 | 70.59 | 27.70 | 49.66 | 4.61 | 62.86 | 43.08 |
| Mag-sp | 4.8% | 1/4 | 57.08 | 18.78 | 2.58 | 0.08 | 12.72 | 18.25 |
| LLM-Pruner | 4.8% | 1/4 | 55.25 | 17.67 | 2.58 | 0.38 | 27.03 | 20.58 |
| Wanda-sp | 4.8% | 1/4 | 45.05 | 13.38 | 0.76 | 0.23 | 3.66 | 12.61 |
| FLAP | 4.8% | 1/4 | 56.44 | 17.84 | 0.83 | 0.00 | 0.14 | 15.05 |
| HARP + FLAP | 4.8% | 1/4 | **60.06** | **21.50** | **31.77** | **1.36** | **30.93** | **29.12** |
| HARP | 3.3% | 1/4 | 61.45 | 21.99 | 36.16 | 1.36 | 34.53 | 31.10 |
| **Mistral-7B-v0.3** | | | | | | | | |
| Dense | 0 | 100% | 69.67 | 28.45 | 36.54 | 2.64 | 58.01 | 39.06 |
| Mag-sp | 4.8% | 1/4 | 63.50 | 25.24 | 7.35 | 0.00 | 14.44 | 22.11 |
| Wanda-sp | 4.8% | 1/4 | 63.15 | 23.16 | 0.76 | 0.00 | 0.02 | 17.42 |
| FLAP | 4.8% | 1/4 | 63.18 | 22.47 | 1.67 | 0.00 | 0.02 | 17.47 |
| HARP + FLAP | 4.8% | 1/4 | **65.64** | **26.40** | **19.71** | **2.64** | **39.98** | **30.88** |
| HARP | 3.6% | 1/4 | 66.19 | 26.90 | 23.88 | 2.49 | 40.55 | 32.00 |
| **Qwen2-7B** | | | | | | | | |
| Dense | 0 | 0 | 61.16 | 26.62 | 78.32 | 20.62 | 59.87 | 49.32 |
| Mag-sp | 1.8% | 1/7 | 54.32 | 20.33 | 52.84 | 6.87 | 36.74 | 34.22 |
| Wanda-sp | 1.8% | 1/7 | 55.73 | 22.33 | 53.37 | 8.53 | 28.26 | 33.64 |
| FLAP | 1.8% | 1/7 | 49.73 | 19.09 | 18.35 | 2.64 | **37.69** | 25.50 |
| HARP + FLAP | 1.8% | 1/7 | **57.76** | **23.46** | **63.91** | **12.61** | 36.20 | **38.79** |
| HARP | 0.9% | 1/7 | 57.61 | 23.27 | 62.24 | 12.16 | 35.63 | 38.18 |
| **Gemma2-9B** | | | | | | | | |
| Dense | 0 | 0 | 71.48 | 31.02 | 68.31 | 14.27 | 70.80 | 51.18 |
| Mag-sp | 3.2% | 1/7 | 64.44 | 22.80 | 0.00 | 0.15 | 3.12 | 18.10 |
| Wanda-sp | 3.2% | 1/7 | 60.29 | 22.05 | 0.00 | 0.00 | 5.58 | 17.58 |
| FLAP | 3.2% | 1/7 | 9.95 | 3.66 | 0.00 | 0.00 | 0.00 | 2.72 |
| HARP + FLAP | 3.2% | 1/7 | **67.57** | **27.81** | **30.63** | **4.61** | **52.97** | **36.72** |
| HARP | 1.6% | 1/7 | 67.69 | 28.42 | 30.02 | 4.00 | 53.02 | 36.63 |

Please note that baseline approaches sometimes apply pruning to either all layers or only a subset of them. To ensure a fair comparison and optimize baseline performance, we also tune the pruning layer range for each baseline model by model based on perplexity, since perplexity is a commonly-used metric for evaluating the performance of pruning algorithms. Details of the experimental setup and the tuning results are provided in Appendix B. All experiments are conducted using NVIDIA H100 80G GPUs.

**Cost analysis and comparison with baselines.** Our implementation performs a *layer-wise* grid search, evaluating $K$ candidate $\alpha$ values per layer using perplexity-based forward passes. Thus, the total cost scales as $L \times K$ forward passes for an $L$-layer model. For the 8-layer setting used in our experiments with $K$=10, this amounts to $8 \times 10 = 80$ forward passes in total. In contrast, inference-only pruning baselines such as Mag-sp, Wanda-sp, and FLAP generally require only a single calibration/inference pass to collect importance statistics, i.e., roughly $\sim 1\times$ the cost of one forward pass. Overall, HARP introduces a one-time search overhead (about $80\times$ more forward passes than single-pass baselines in this 8-layer case), but remains substantially cheaper than training-based methods; moreover, this cost is amortized since the search is performed only once to select a good $\alpha$.

**Benchmarks.** For our evaluation, we employ the widely-used `lm-evaluation-harness` package (Gao et al., 2024) to conduct experiments on both generation and discriminative tasks. Our generation tasks include 5-shot TriviaQA (Joshi et al., 2017), 5-shot NaturalQuestions (Kwiatkowski et al., 2019), 5-shot GSM8K (Cobbe et al., 2021), 4-shot MATH-hard (Hendrycks et al., 2021b; Lewkowycz et al., 2022), 3-shot COT (Wei et al., 2022a), and BBH (Suzgun et al., 2023). For discriminative tasks, we use 5-shot WinoGrande (Sakaguchi et al., 2021), 25-shot ARC-Challenge (Clark et al., 2018), 0-shot BoolQ (Clark et al., 2019), 0-shot OpenBookQA (Mihaylov et al., 2018), 0-shot PIQA (Bisk et al., 2020), and 5-shot MMLU (Hendrycks et al., 2021a). We report the accuracy for these tasks as recommended by the `lm-evaluation-harness` package. We also evaluate

Table 3: Performance on discriminative tasks across LLaMA3.1-8B, Mistral-7B-v0.3, Qwen2-7B, and Gemma2-9B (%). Best is **bold**

| Method | Attention Head Pruning Ratio | WinoGrande | ARC-Challenge | BoolQ | OpenBookQA | PIQA | MMLU | Average |
|---|---|---|---|---|---|---|---|---|
| **LLaMA3.1-8B** | | | | | | | | |
| Dense | 0 | 77.58 | 54.86 | 82.05 | 33.40 | 80.09 | 65.30 | 65.55 |
| Mag-sp | 1/4 | **77.43** | 49.15 | 78.26 | 31.80 | 78.02 | 58.20 | 62.14 |
| LLM-Pruner | 1/4 | 77.03 | 49.83 | 79.14 | **33.00** | **78.94** | 34.33 | 58.71 |
| Wanda-sp | 1/4 | 76.64 | 45.99 | **80.76** | 28.40 | 76.93 | 39.45 | 58.03 |
| FLAP | 1/4 | 65.51 | 40.27 | 72.97 | 31.60 | 77.97 | 32.01 | 53.39 |
| HARP + FLAP | 1/4 | 76.64 | **51.02** | 79.82 | 31.80 | 78.89 | **61.96** | **63.35** |
| HARP | 1/4 | 78.06 | 53.75 | 75.02 | 33.20 | 78.89 | 64.83 | 63.96 |
| **Mistral-7B-v0.3** | | | | | | | | |
| Dense | 0 | 78.45 | 57.76 | 82.11 | 33.20 | 80.14 | 62.39 | 65.68 |
| Mag-sp | 1/4 | 76.48 | 53.33 | 69.48 | 31.40 | **79.65** | 57.04 | 61.23 |
| Wanda-sp | 1/4 | 68.67 | 48.81 | 71.31 | 31.80 | **79.65** | 33.51 | 55.62 |
| FLAP | 1/4 | 62.43 | 41.98 | 65.57 | 29.20 | 78.99 | 39.46 | 52.94 |
| HARP + FLAP | 1/4 | **77.58** | **53.92** | **81.13** | **33.60** | 79.54 | **60.68** | **64.41** |
| HARP | 1/4 | 77.90 | 55.55 | 80.49 | 33.20 | 79.22 | 61.52 | 64.64 |
| **Qwen2-7B** | | | | | | | | |
| Dense | 0 | 77.03 | 58.11 | 84.80 | 35.20 | 79.92 | 70.34 | 67.57 |
| Mag-sp | 1/7 | 68.90 | 47.53 | 77.95 | 30.20 | 77.37 | 59.24 | 60.20 |
| Wanda-sp | 1/7 | 63.30 | 53.92 | 81.19 | 31.00 | 79.05 | 62.23 | 61.78 |
| FLAP | 1/7 | 74.35 | 49.23 | 82.78 | 30.00 | 76.61 | 66.67 | 63.27 |
| HARP + FLAP | 1/7 | **75.61** | **57.08** | **85.23** | **35.00** | **79.33** | **67.97** | **66.70** |
| HARP | 1/7 | 76.09 | 57.59 | 84.98 | 34.60 | 79.38 | 68.07 | 66.79 |
| **Gemma2-9B** | | | | | | | | |
| Dense | 0 | 79.87 | 64.76 | 84.19 | 33.80 | 81.34 | 70.64 | 69.10 |
| Mag-sp | 1/7 | 66.22 | 58.19 | 79.91 | 34.00 | 80.30 | 53.56 | 62.03 |
| Wanda-sp | 1/7 | **79.64** | 65.02 | **84.13** | **36.20** | **81.39** | **70.60** | **69.50** |
| FLAP | 1/7 | 76.09 | 53.41 | 78.44 | 29.00 | 78.56 | 67.82 | 63.89 |
| HARP + FLAP | 1/7 | 79.16 | **65.44** | 83.49 | 35.20 | 80.36 | 70.27 | 68.99 |
| HARP | 1/7 | 79.87 | 64.85 | 83.00 | 34.60 | 80.96 | 70.60 | 68.98 |

long-context generation performance on the LongBench benchmark (Bai et al., 2024), which includes 16 datasets. Details about benchmarks can be found in Appendix C.

## 3.2 Main Results

**Generation Tasks.** Table 2 summarizes the results of all pruning algorithms on 5 generation tasks, showing the superior performance of our proposed method, HARP, across a range of generation tasks and language models. Our method consistently outperforms all four baselines across different models and benchmarks.

Further analysis reveals a notable performance gap between GSM8K and TriviaQA. This difference can be largely attributed to the varying response lengths required by each task. For TriviaQA, unpruned LLMs typically encounter the end-of-sequence (EOS) token within just 16 tokens. In contrast, GSM8K often requires up to 256 tokens to generate a complete solution. Since pruned LLMs generate output one token at a time, they are more vulnerable to error propagation, especially in tasks that demand longer responses. Each additional token introduces more opportunity for cumulative errors. As a result, the extended response length required for GSM8K leads to a more significant performance drop in pruned models, compared to the relatively short responses needed for TriviaQA.

**Discriminative Tasks.** Table 3 presents the performance of four LLMs on discriminative tasks, along with the average results across six benchmarks. The results show that our pruning algorithm achieves the highest average performance on LLaMA3.1-8B, Mistral-8B-v0.3, and Qwen2-7B, and performs on par with the best baseline on Gemma2-9B. Interestingly, the performance drop on discriminative tasks is less pronounced compared to generation tasks. This can be attributed to the nature of discriminative tasks, which typically

Table 4: Performance on long-context generation tasks across LLaMA3.1-8B and Mistral-7B-v0.3. Best is **bold**

| | | Long-Context Generation Task Performance Evaluation | | | | | | | | | | | | | | | | |
|---|---|---|---|---|---|---|---|---|---|---|---|---|---|---|---|---|---|---|
| | | Single-Document QA | | | Multi-Document QA | | | Summarization | | | Few-shot Learning | | | Synthetic | | Code | |
| Model | Method | NrtvQA | Qasper | MF-en | HotpotQA | 2WikiMQA | Musique | GovReport | QMSum | MultiNews | TREC | TriviaQA | SAMSum | PCount | PRe | Lcc | RB-P |
| LLaMA3.1-8B | Dense | 20.32 | 11.53 | 32.53 | 12.10 | 14.51 | 8.22 | 29.20 | 23.11 | 2.35 | 68.50 | 90.49 | 46.45 | 3.62 | 19.50 | 68.70 | 62.16 |
| | Mag-sp | 11.31 | 8.97 | **23.98** | 7.34 | 7.75 | 6.22 | **16.44** | 19.75 | 2.96 | 58.00 | 80.02 | 33.17 | 3.75 | 8.25 | 36.72 | **39.16** |
| | LLM-Pruner | 9.68 | 6.71 | 19.36 | 6.46 | 5.68 | 4.51 | 13.03 | 20.75 | 0.23 | 59.50 | 73.36 | **36.18** | 4.00 | 3.70 | 36.67 | 38.81 |
| | Wanda-sp | **14.05** | 5.98 | 17.48 | 6.90 | 6.27 | 3.83 | 8.28 | 19.05 | 0.48 | 56.00 | 68.25 | 29.84 | **4.70** | 6.12 | 28.66 | 30.04 |
| | FLAP | 7.86 | 5.14 | 13.12 | 6.46 | 5.50 | 4.37 | 8.39 | 8.31 | 2.87 | 15.50 | 73.15 | 10.06 | 1.55 | 4.01 | 33.65 | 34.57 |
| | HARP+FLAP | 12.68 | **11.52** | 19.96 | **10.15** | **10.02** | **6.40** | 13.10 | **20.97** | 0.00 | **63.50** | 81.04 | 24.49 | 1.50 | **10.83** | 17.13 | 31.42 |
| | HARP | 13.21 | 10.20 | 23.98 | 10.21 | 10.31 | 6.68 | 19.19 | 19.50 | 0.00 | 64.00 | 82.57 | 25.18 | 6.00 | 18.38 | 14.47 | 25.19 |
| Mistral-7B-v0.3 | Dense | 15.21 | 6.33 | 28.32 | 10.59 | 11.06 | 5.15 | 27.61 | 21.36 | 25.88 | 72.00 | 90.26 | 45.89 | 1.00 | 12.00 | 64.85 | 57.16 |
| | Mag-sp | **12.34** | 5.25 | **22.64** | 9.14 | 8.75 | 5.20 | 9.26 | 18.97 | 6.61 | **61.50** | 80.96 | 31.70 | 0.62 | **4.67** | 49.68 | **39.31** |
| | Wanda-sp | 2.37 | 4.78 | 6.98 | 7.18 | 3.65 | 3.89 | 7.41 | 14.19 | 8.54 | 27.50 | 63.93 | 20.25 | **2.83** | 3.29 | 39.46 | 37.05 |
| | FLAP | 1.93 | 4.48 | 10.21 | 9.08 | 5.85 | 3.50 | 9.33 | 16.91 | 13.64 | 28.00 | 72.94 | 9.21 | 2.50 | 4.25 | 45.61 | 36.63 |
| | HARP+FLAP | 10.27 | **7.08** | 21.09 | **9.97** | **9.94** | **5.67** | **21.89** | 19.70 | 21.04 | 56.50 | **88.71** | **32.76** | 0.50 | 4.38 | 29.03 | 37.21 |
| | HARP | 12.79 | 8.87 | 22.42 | 10.40 | 10.49 | 5.91 | 23.32 | 19.78 | 23.25 | 57.00 | 88.55 | 32.73 | 1.00 | 8.71 | 40.22 | 43.19 |

Table 5: Ablation study on layer selection strategies for pruning attention heads, evaluated on six discriminative tasks using LLaMA3.1-8B (%).

| LLaMA3.1-8B | | | | | | | |
|---|---|---|---|---|---|---|---|
| Method | WinoGrande | ARC-Challenge | BoolQ | OpenBookQA | PIQA | MMLU | Average |
| Bottom-8 | 50.43 | 19.20 | 37.83 | 13.40 | 52.50 | 22.95 | 32.72 |
| Hessian-8 | 52.88 | 22.35 | 62.26 | 19.20 | 59.09 | 23.01 | 39.80 |
| Similarity-8 | 72.30 | 43.94 | 79.91 | 25.80 | 72.25 | 57.53 | 58.62 |
| Top-8 | **72.53** | **46.59** | **82.14** | **28.20** | **74.16** | **58.73** | **60.39** |

involve selecting a single correct answer from a small set of options. Such tasks are generally less difficult than generation tasks, which require predicting the next token from the full vocabulary, making them more sensitive to pruning.

## 3.3 Long-Context Modeling

In this section, we evaluate the performance of structured pruning algorithms on long-context generation tasks. We use LLaMA3.1-8B and Mistral-8B-v0.3 for our experiments. For evaluation, we use LongBench (Bai et al., 2024), which encompasses a diverse range of tasks categorized as follows: single-document QA (NrtvQA, Qasper, MF-en), multi-document QA (HotpotQA, 2WikiMQA, Musique), summarization (GovReport, QMSum, MultiNews), few-shot learning (TREC, TriviaQA, SAMSum), synthetic tasks (PCount, PRe), and code-related tasks (Lcc, RB-P). This comprehensive benchmark allows us to assess performance across various long-context scenarios.

Table 4 demonstrates the performance of various pruning algorithms across long-context benchmarks. Our proposed method outperforms baseline approaches in most benchmarks on both LLaMA3.1-8B and Mistral-7B-v0.3 architectures. The improvements are particularly significant in multi-document QA tasks such as HotpotQA and 2WikiMQA, indicating better preservation of reasoning abilities and long-range dependencies. While other methods like Mag-SP achieve strong results in specific areas (e.g., MF-en), our method attains the highest number of rank-1 performances across diverse benchmarks, highlighting its robustness for long-context processing.

## 3.4 Exploring Layer Selection for Pruning Attention Heads

In this section, we conduct an ablation study to support our claim that attention heads in higher layers are less important than those in lower layers. We evaluate four different strategies for selecting layers to prune, using LLaMA3.1-8B for our experiments. The first strategy prunes attention heads in the lowest 8 layers

Table 6: Ablation study of HARP on 11 tasks using LLaMA3.1-8B, comparing performance with and without $\alpha$ search (%). Best is **bold**

| | | Discriminative Tasks | | | | | | |
|---|---|---|---|---|---|---|---|---|
| Method | $\alpha$ | WinoGrande | ARC-Challenge | BoolQ | OpenBookQA | PIQA | MMLU | Average |
| HARP | 1.0 | 72.53 | 46.59 | **82.14** | 28.20 | 74.16 | 58.73 | 60.39 |
| HARP | searched | **78.06** | **53.75** | 75.02 | **33.20** | **78.89** | **64.83** | **63.96** |
| | | Short-Context Generation Tasks | | | | | | |
| Method | $\alpha$ | TriviaQA | NaturalQuestions | GSM8K | MATH-hard | BBH | Average | |
| HARP | 1.0 | 12.43 | 3.24 | 1.29 | 0.00 | 0.03 | 3.40 | |
| HARP | searched | **61.45** | **21.99** | **36.16** | **1.36** | **34.53** | **31.10** | |

**(Bottom-8)**. The second selects 8 layers using a Hessian-based metric proposed in (Yang et al., 2023a) **(Hessian-8)**. The third leverages a similarity-based metric for the self-attention module, as proposed in (He et al., 2024), to select 8 layers **(Similarity-8)**. The fourth strategy, which is our approach, prunes attention heads in the highest 8 layers **(Top-8)**. We set $\alpha = 1$ for all layers. Additional details on the Hessian-based and similarity-based metrics are provided in Appendix D.

For both the Hessian-8 and Similarity-8 strategies, we compute the respective metrics using the WikiText dataset (Merity et al., 2017). The selected layer indices for pruning are [12, 13, 14, 15, 19, 22, 24, 26] based on the Hessian-based metric, and [23, 24, 25, 26, 27, 28, 29, 30] based on the similarity-based metric. Based on the selected layer indices, we observe that the Hessian-8 strategy prunes attention heads in the middle 8 layers, the Similarity-8 strategy targets layers 23 to 30, and the Top-8 strategy focuses on layers 24 to 31. Table 5 reports the results of these four strategies across six discriminative tasks. The results indicate that pruning attention heads in the higher layers yields better performance than pruning those in the lower layers. These findings support our theoretical claim that attention heads in higher layers are less important to overall model performance.

### 3.5 $\alpha$ Search for Rescaling

To determine the optimal values of $\alpha$, we compute perplexity on the WikiText (Merity et al., 2017) and Pile10K (Nanda, 2022) datasets using LLaMA3.1-8B. The resulting optimal $\alpha$ values are $[0.8, 0.2, 0.1, 0.1, 0.0, 0.1, 0.0, 0.0]$ for WikiText and $[0.8, 0.2, 0.2, 0.1, 0.0, 0.1, 0.1, 0.0]$ for Pile10K, with indices corresponding to layers counted from the top. The similarity between the two sets of values suggests a consistent trend in the layer-wise importance.

We further evaluate both $\alpha$ configurations by measuring perplexity on both datasets. Using the values obtained from WikiText, the perplexity scores are 12.16 on WikiText and 28.94 on Pile10K. Conversely, using the values derived from Pile10K yields scores of 12.23 on WikiText and 28.50 on Pile10K. These results demonstrate that both configurations produce comparable perplexity on each dataset, confirming the robustness of the searched $\alpha$ values. For simplicity, we adopt the $\alpha$ values derived from WikiText throughout this work. Figure 2 presents the searched $\alpha$ values for LLaMA3.1-8B, Mistral-7B-v0.3, Qwen2-7B, and Gemma2-9B. Our results show that the optimal $\alpha$ values vary across models and pruned layers. For LLaMA3.1-8B, Mistral-7B-v0.3, and Qwen2-7B, we observe a general trend where the searched $\alpha$ values increase with the layer index. Even for models with the same architecture but different sizes, such as LLaMA3.1-8B and LLaMA3.1-70B, the optimal $\alpha$ values can be different. For LLaMA3.1-70B, the searched $\alpha$ values are $[0.0, 0.1, 0.0, 0.0, 0.2, 0.0, 0.0, 0.0]$, which is noticeably different from those obtained for LLaMA3.1-8B.

Please note that the WikiText dataset provided by `lm-evaluation-harness` includes only 62 sub-tasks. For LLaMA3.1-8B, our search involves 8 separate for-loops, each sweeping $\alpha$ values from 0 to 1.0. As a result, the total number of evaluations amounts to 5,456 sub-tasks. We conduct the experiments using an NVIDIA H100 GPU, and the total runtime remains under two hours, which is comparable to the time required for a typical experiment on a generation task.

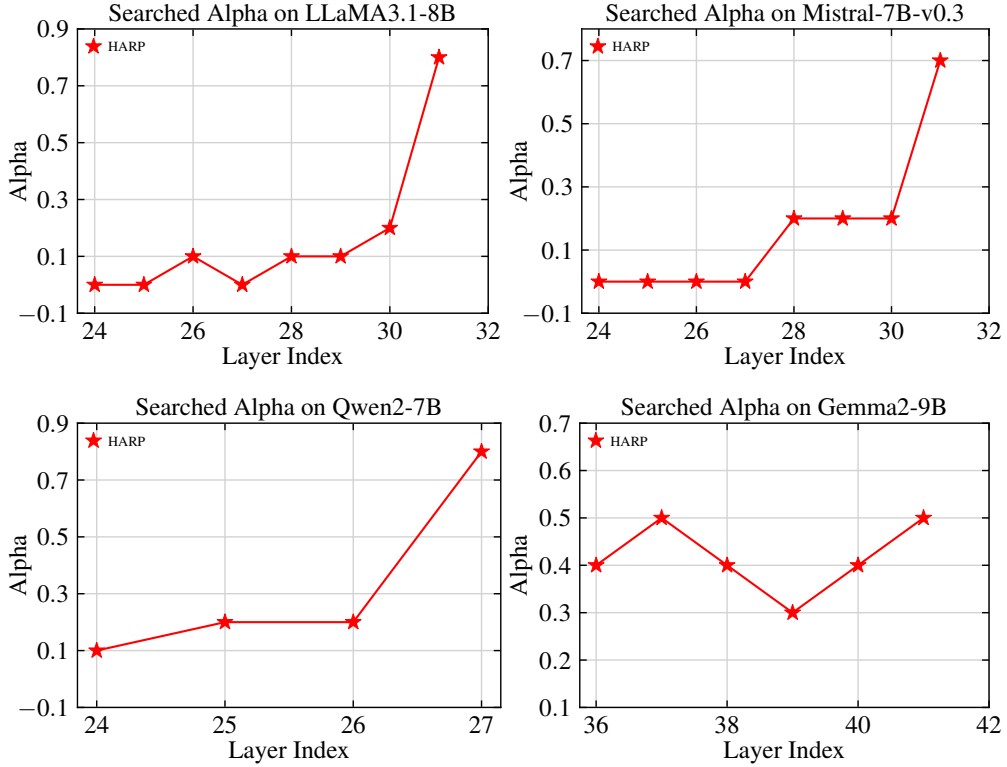

Figure 2: Searched alpha on LLaMA3.1-8B, Mistral-7B-v0.3, Qwen2-7B and Gemma2-9B.

**Ablation Study.** We conduct an ablation study to demonstrate the effectiveness of $\alpha$ search for rescaling. As shown in the Table 6, incorporating $\alpha$ search consistently improves performance, particularly on generation tasks, where we observe a substantial performance gain.

### 3.6 Efficiency Analysis

In this section, we analyze the time complexity of our pruned model. We conduct experiments with varying sequence lengths to measure inference processing time. Using LLaMA3.1-8B as our test model, we evaluate sequences of lengths 1024, 2048, 4096, 8192, 16384, 32768, and 65536. All measurements are conducted on a single NVIDIA H100 80G GPU with each experiment repeated 10 times to ensure statistical reliability. For comparison purposes, we test both our pruned model (with 8 layer attention heads pruned) and the original model (with 0 attention heads pruned) to quantify the efficiency gains. Our measurements reveal consistent performance across all runs, with tight 95% confidence intervals: for the original model, inference times ranged from $0.0371 \pm 0.0001$s for 1024 tokens to $5.4466 \pm 0.0022$s for 65536 tokens; for the pruned model, times ranged from $0.0355 \pm 0.0001$s for 1024 tokens to $4.5394 \pm 0.0063$s for 65536 tokens. Figure 3 demonstrates the significant speed gains achieved by our pruned model. As sequence length increases, our speed improvement becomes more substantial. With just a 3.3% parameter reduction, we achieve a 16.7% improvement in processing speed for 65,536-token sequences. This highlights how targeted attention head pruning can substantially reduce inference latency in long-context tasks.

## 4 Related Work

**Pruning.** Pruning is a widely adopted and efficient technique in both Computer Vision and Large Language Models. It can be categorized into two main types: structured pruning and unstructured pruning. structured pruning (Lagunas et al., 2021; Xia et al., 2022; Kurtic et al., 2023; He & Xiao, 2023; Xia et al., 2024) involves removing entire filters from neural networks, making it particularly conducive to model deployment. On

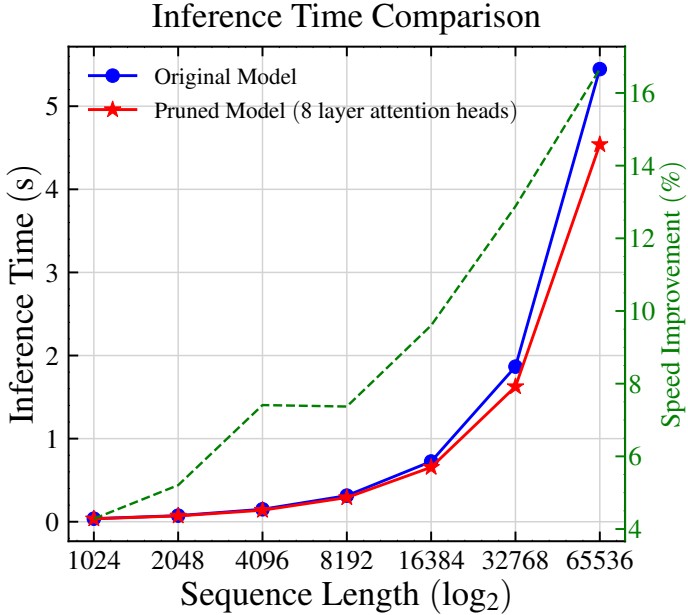

Figure 3: Time complexity and runtime scaling with sequence length for the original and pruned models. Solid lines show the absolute runtime (left y-axis), while the green dashed line denotes the relative speedup (%) of the pruned model over the original, plotted on the secondary y-axis (right).

the other hand, unstructured pruning (Chen et al., 2020; Sanh et al., 2020) focuses on removing individual neurons within the network. Some recent works (Men et al., 2024; Zhong et al., 2024; Gromov et al., 2024; He et al., 2024; Siddiqui et al., 2024; Liu et al., 2024b; Zhang et al., 2024a; Jaiswal et al., 2024b; Chen et al., 2024b; Kim et al., 2024a) have been proposed to prune blocks in the higher layers of LLMs.

**Quantization.** Quantization approaches (Wei et al., 2022b; Yao et al., 2022; Frantar et al., 2023; Xiao et al., 2023; Dettmers & Zettlemoyer, 2023; Park et al., 2024; Lin et al., 2024) compress language models by converting weights and activations to lower precision formats such as 8-bit, 4-bit, or even 2-bit integers. This reduction in precision substantially decreases memory requirements and enhances inference speed, enabling the deployment of large language models in environments with limited computational resources.

**KV Cache Compression.** Recent works (Liu et al., 2023; Anagnostidis et al., 2023; Zhang et al., 2023; Ge et al., 2024; Xiao et al., 2024; Kim et al., 2024b; Zhang et al., 2024c; Nawrot et al., 2024; Tang et al., 2024; Liu et al., 2024d; Dong et al., 2024a; Yue et al., 2024; Cai et al., 2024; Liu et al., 2024c;a; Hooper et al., 2024; Sun et al., 2024c; Chen et al., 2024a; Sun et al., 2024a; Jiang et al., 2024; Li et al., 2024) have focused on compressing the KV cache to reduce GPU memory footprint. Some (Liu et al., 2024d; Hooper et al., 2024; Yue et al., 2024) of these methods utilize quantization techniques to achieve compression. Additionally, approaches like H2O (Zhang et al., 2023) evict less important tokens, while LESS (Dong et al., 2024a) generates condensed representations through training to optimize memory usage.

**Low-rank Approximation.** Low-rank approximation methods (Hu et al., 2022; Dettmers et al., 2023; Lin et al., 2025; Wang et al., 2025; Chang et al., 2025) project token representations into a low-dimensional latent space and subsequently reconstruct the full representations via up-projection. This class of methods significantly reduces the number of trainable parameters during fine-tuning.

## 5 Conclusion

In this work, we propose HARP, a novel pruning algorithm that removes attention heads in the higher layers of LLMs. Extensive experiments across short-context and long-context generation tasks, as well as

discriminative benchmarks, demonstrate the superiority of our proposed method. Ablation studies further reveal the importance of incorporating a rescaling parameter to maintain performance.

Beyond these empirical gains, HARP has clear practical implications: by reducing inference cost without requiring architectural changes or retraining from scratch, it can facilitate the deployment of LLMs in resource-constrained environments, enable faster serving for interactive applications, and make long-context reasoning more affordable in real-world systems.

This study has limitations: we mainly focus on attention-head pruning in higher layers, leaving other components underexplored. Future work includes extending HARP to larger and multimodal models, combining it with techniques such as quantization or KV-cache compression, and developing more task-aware pruning criteria to further improve robustness and efficiency.

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

# A Theoretical Analysis

**Definition 2.** *(Shi et al., 2022) Define the* over-smoothing subspace

$$\mathcal{M} := \left\{ \mathbf{Y} \in \mathbb{R}^{N \times d} \,\middle|\, \mathbf{Y} = \mathbf{e}\mathbf{C}, \ \mathbf{C} \in \mathbb{R}^{1 \times d} \right\},$$

*where* $\mathbf{e} = [1, 1, \ldots, 1]^\top \in \mathbb{R}^{N \times 1}$, $N$ *is the number of tokens, and* $d$ *is the dimension of token representations.*

Intuitively, any matrix $\mathbf{Y} \in \mathcal{M}$ has *identical rows*: every token in the sequence shares exactly the same representation vector. We therefore regard $\mathcal{M}$ as the set of "maximally over-smoothed" token representations, where no token-specific information is preserved.

**Definition 3.** *(Shi et al., 2022) Define the distance between a matrix* $\mathbf{H} \in \mathbb{R}^{N \times d}$ *and the over-smoothing subspace* $\mathcal{M}$ *as*

$$d_{\mathcal{M}}(\mathbf{H}) := \min_{\mathbf{Y} \in \mathcal{M}} \|\mathbf{H} - \mathbf{Y}\|_F,$$

*where* $\| \cdot \|_F$ *denotes the Frobenius norm.*

**Lemma 1** ((Shi et al., 2022)). *For a self-attention matrix* $\hat{\mathbf{A}}$, *any* $\mathbf{H}, \mathbf{B} \in \mathbb{R}^{N \times d}$ *and any* $\alpha_1, \alpha_2 \geq 0$, *we have:*

$$d_{\mathcal{M}}(\mathbf{H}\mathbf{W}) \leq s\, d_{\mathcal{M}}(\mathbf{H}), \qquad\qquad d_{\mathcal{M}}(\sigma(\mathbf{H})) \leq d_{\mathcal{M}}(\mathbf{H}), \qquad (4)$$

$$d_{\mathcal{M}}(\alpha_1 \mathbf{H} + \alpha_2 \mathbf{B}) \leq \alpha_1 d_{\mathcal{M}}(\mathbf{H}) + \alpha_2 d_{\mathcal{M}}(\mathbf{B}), \qquad d_{\mathcal{M}}(\hat{\mathbf{A}}\mathbf{H}) \leq \sqrt{\lambda_{\max}}\, d_{\mathcal{M}}(\mathbf{H}), \qquad (5)$$

*where* $\lambda_{\max}$ *is the largest eigenvalue of* $\hat{\mathbf{A}}^\top(\mathbf{I} - \mathbf{e}\mathbf{e}^\top)\hat{\mathbf{A}}$ *and* $s$ *is the largest singular value of* $\mathbf{W}$.

By Lemma 1, we have the following result.

**Lemma 2** ((Shi et al., 2022)). *For a Transformer block with* $h$ *heads, we have*

$$d_{\mathcal{M}}(\mathbf{H}^{(\ell+1)}) \leq v\, d_{\mathcal{M}}(\mathbf{H}^{(\ell)}), \qquad (6)$$

*where* $v = (1 + s^2)(1 + \sqrt{\lambda}hs)/(\beta_1\beta_2)$, $s > 0$ *is the largest element among all singular values of all* $\mathbf{W}^{(\ell)}$, $\lambda$ *is the largest eigenvalue of all* $\hat{\mathbf{A}}^\top(\mathbf{I} - \mathbf{e}\mathbf{e}^\top)\hat{\mathbf{A}}$ *for each self-attention matrix* $\hat{\mathbf{A}}$, *and* $\beta_1$, $\beta_2$ *are the minimum standard deviations for the two layer normalization operations.*

The proofs of these two lemmas can be found in (Shi et al., 2022). Lemma 2 shows that if $v < 1$, then $d_{\mathcal{M}}(\mathbf{H}^{(\ell+1)}) < d_{\mathcal{M}}(\mathbf{H}^{(\ell)})$, i.e., the hidden representations move closer to $\mathcal{M}$ as the depth $\ell$ increases. In other words, deeper layers become progressively more over-smoothed.

**Definition 1.** *(Average pairwise cosine similarity) Define the average pairwise cosine similarity among tokens in a matrix* $\mathbf{H} \in \mathbb{R}^{N \times d}$ *as*

$$\mathrm{Sim}(\mathbf{H}) := \frac{2}{N(N-1)} \sum_{1 \leq i < j \leq N} \frac{\mathbf{h}_i^\top \mathbf{h}_j}{\|\mathbf{h}_i\|_2 \|\mathbf{h}_j\|_2}. \qquad (7)$$

*A value* $\mathrm{Sim}(\mathbf{H}) \approx 1$ *indicates that all token vectors are nearly parallel (up to scaling), while smaller values correspond to more diverse token directions. Thus,* $\mathrm{Sim}(\mathbf{H})$ *measures the global similarity of token representations.*

**Lemma 3.** *For any token representation matrix* $\mathbf{H} \in \mathbb{R}^{N \times d}$, *if its distance to the over-smoothing subspace* $\mathcal{M}$ *tends to zero, then its average pairwise similarity tends to one, i.e.,*

$$d_{\mathcal{M}}(\mathbf{H}) \to 0 \quad \implies \quad \mathrm{Sim}(\mathbf{H}) \to 1.$$

*Proof.* This is the proof of Lemma 3. By definition, $\mathcal{M} = \{\mathbf{Y} \in \mathbb{R}^{N \times d} \mid \mathbf{Y} = \mathbf{e}\mathbf{C}, \mathbf{C} \in \mathbb{R}^{1 \times d}\}$, where all token embeddings are equal (each row of $\mathbf{Y}$ is $\mathbf{C}$). If $d_{\mathcal{M}}(\mathbf{H}) \to 0$, then there exists some $\mathbf{C}$ such that $\mathbf{H} \to \mathbf{e}\mathbf{C}$, meaning $\mathbf{h}_i \to \mathbf{C}$ for all $i$. Hence,

$$\mathbf{h}_i^\top \mathbf{h}_j \to \|\mathbf{C}\|_2^2 \quad \text{and} \quad \|\mathbf{h}_i\|_2 \|\mathbf{h}_j\|_2 \to \|\mathbf{C}\|_2^2,$$

so for any $i \neq j$ we have

$$\frac{\mathbf{h}_i^\top \mathbf{h}_j}{\|\mathbf{h}_i\|_2 \|\mathbf{h}_j\|_2} \to 1.$$

Averaging over all pairs $(i, j)$ then yields $\mathrm{Sim}(\mathbf{H}) \to 1$. $\qquad\square$

**Theorem 1.** *When* $\mathrm{Sim}(\mathbf{H}^{(\ell)}) \to 1$, *token representations in layer $\ell$ become nearly parallel. Since queries and keys are linear projections of* $\mathbf{H}^{(\ell)}$, *their dot products across positions become almost constant, so the row-wise softmax yields nearly uniform attention weights. The head thus degenerates into averaging value vectors and cannot distinguish tokens. Conversely, if* $\mathrm{Sim}(\mathbf{H}^{(\ell)})$ *is bounded away from 1, attention remains non-uniform and can model meaningful token-to-token dependencies.*

*Proof.* This is the proof of Theorem 1. Suppose $\mathrm{Sim}(\mathbf{H}^{(\ell)}) \to 1$. By Lemma 3, this implies $d_{\mathcal{M}}(\mathbf{H}^{(\ell)}) \to 0$, so all token embeddings $\mathbf{h}_i^{(\ell)}$ converge to a shared direction, i.e., they become almost parallel.

In a self-attention layer, the query and key matrices are obtained from $\mathbf{H}^{(\ell)}$ by linear projections:

$$\mathbf{Q}^{(\ell)} = \mathbf{H}^{(\ell)} \mathbf{W}_Q, \qquad \mathbf{K}^{(\ell)} = \mathbf{H}^{(\ell)} \mathbf{W}_K.$$

Since linear maps preserve the property of vectors being almost parallel, the rows of $\mathbf{Q}^{(\ell)}$ and $\mathbf{K}^{(\ell)}$ also become nearly parallel when $\mathrm{Sim}(\mathbf{H}^{(\ell)}) \to 1$. Consequently, the inner products $\mathbf{q}_i^\top \mathbf{k}_j$ for all $i, j$ converge to (almost) the same value:

$$\mathbf{q}_i^\top \mathbf{k}_j \approx c \quad \text{for all } i, j.$$

The attention weights for a fixed query position $i$ are given by

$$a_{ij} = \frac{\exp(\mathbf{q}_i^\top \mathbf{k}_j / \sqrt{d_k})}{\sum_{j'} \exp(\mathbf{q}_i^\top \mathbf{k}_{j'} / \sqrt{d_k})}.$$

If all logits $\mathbf{q}_i^\top \mathbf{k}_j$ are (approximately) equal, then the softmax output $[a_{i1}, \dots, a_{iN}]$ approaches the uniform distribution over $\{1, \dots, N\}$. Therefore, the attention weights can no longer distinguish between different tokens, and the corresponding attention head effectively reduces to averaging the value vectors across positions.

As a result, when $\mathrm{Sim}(\mathbf{H}^{(\ell)})$ is close to 1 in high layers, self-attention behaves like a simple smoothing operator that mixes already similar representations, contributing little to further feature refinement or discrimination. In contrast, in lower layers where $\mathrm{Sim}(\mathbf{H}^{(\ell)})$ is bounded away from 1, attention weights remain non-uniform and can capture meaningful token-to-token dependencies, making those layers more informative. $\qquad\square$

**Proof of Proposition 1.**

*Proof.* Recall that $\hat{\mathbf{A}} \in \mathbb{R}^{N \times N}$ is a row-normalized attention matrix and

$$\mathbf{H}' = \hat{\mathbf{A}}\mathbf{H}, \quad \mathbf{H} \in \mathbb{R}^{N \times d}.$$

We first show that if every row of $\hat{\mathbf{A}}$ is a one-hot vector (i.e., has exactly one entry equal to 1 and all others equal to 0), then the Frobenius norm is preserved for any $\mathbf{H}$. In this case, $\hat{\mathbf{A}}$ is a row-permutation matrix: there exists a permutation $\pi$ of $\{1, \dots, N\}$ such that

$$(\hat{\mathbf{A}}\mathbf{H})_{i,:} = \mathbf{H}_{\pi(i),:}, \quad \forall i.$$

Thus $\hat{\mathbf{A}}$ only permutes the rows of $\mathbf{H}$, and we have

$$\|\hat{\mathbf{A}}\mathbf{H}\|_F^2 = \sum_{i=1}^N \sum_{j=1}^d \left((\hat{\mathbf{A}}\mathbf{H})_{ij}\right)^2 = \sum_{i=1}^N \sum_{j=1}^d \left(\mathbf{H}_{\pi(i),j}\right)^2 = \sum_{i=1}^N \sum_{j=1}^d \mathbf{H}_{ij}^2 = \|\mathbf{H}\|_F^2.$$

Therefore, if each row of $\hat{\mathbf{A}}$ is one-hot, then $\|\mathbf{H}'\|_F = \|\mathbf{H}\|_F$ for all $\mathbf{H}$.

We now prove the converse: if a row-normalized attention matrix $\hat{\mathbf{A}}$ preserves the Frobenius norm for all inputs, then each row of $\hat{\mathbf{A}}$ must be one-hot. Assume that

$$\|\hat{\mathbf{A}}\mathbf{H}\|_F = \|\mathbf{H}\|_F \quad \text{for all } \mathbf{H} \in \mathbb{R}^{N \times d}.$$

Consider the special case where $\mathbf{H}$ has only one nonzero column: let $\mathbf{x} \in \mathbb{R}^N$ be arbitrary and define

$$\mathbf{H} = \begin{bmatrix} \mathbf{x} & \mathbf{0} & \cdots & \mathbf{0} \end{bmatrix} \in \mathbb{R}^{N \times d}.$$

Then

$$\hat{\mathbf{A}}\mathbf{H} = \begin{bmatrix} \hat{\mathbf{A}}\mathbf{x} & \mathbf{0} & \cdots & \mathbf{0} \end{bmatrix},$$

and the Frobenius norm condition reduces to

$$\|\hat{\mathbf{A}}\mathbf{x}\|_2 = \|\mathbf{x}\|_2 \quad \text{for all } \mathbf{x} \in \mathbb{R}^N.$$

Hence $\hat{\mathbf{A}}$ is an isometry on $\mathbb{R}^N$, which implies

$$\mathbf{x}^\top \hat{\mathbf{A}}^\top \hat{\mathbf{A}}\mathbf{x} = \|\hat{\mathbf{A}}\mathbf{x}\|_2^2 = \|\mathbf{x}\|_2^2 = \mathbf{x}^\top \mathbf{I}\mathbf{x} \quad \forall \mathbf{x},$$

so $\hat{\mathbf{A}}^\top \hat{\mathbf{A}} = \mathbf{I}$. Thus $\hat{\mathbf{A}}$ is an orthogonal matrix.

On the other hand, as an attention matrix, $\hat{\mathbf{A}}$ is row-normalized with nonnegative entries:

$$\hat{\mathbf{A}}_{ij} \geq 0, \quad \sum_{j=1}^{N} \hat{\mathbf{A}}_{ij} = 1 \quad \text{for each row } i.$$

Let $\mathbf{a}_i$ denote the $i$-th row of $\hat{\mathbf{A}}$. Orthogonality implies $\|\mathbf{a}_i\|_2 = 1$ for all $i$, while row-normalization gives $\sum_j \mathbf{a}_{ij} = 1$ and $\mathbf{a}_{ij} \geq 0$. Therefore,

$$\sum_{j=1}^{N} \mathbf{a}_{ij} = 1, \quad \sum_{j=1}^{N} \mathbf{a}_{ij}^2 = 1, \quad \mathbf{a}_{ij} \geq 0.$$

For any $0 < a < 1$, we have $a^2 < a$. Hence, if any entry of $\mathbf{a}_i$ satisfies $0 < \mathbf{a}_{ij} < 1$, then

$$\sum_{j=1}^{N} \mathbf{a}_{ij}^2 < \sum_{j=1}^{N} \mathbf{a}_{ij} = 1,$$

which contradicts $\sum_j \mathbf{a}_{ij}^2 = 1$. Thus every entry of $\mathbf{a}_i$ must belong to $\{0, 1\}$. Since the entries are nonnegative and the row sum is 1, each row has exactly one entry equal to 1 and all others equal to 0, i.e., each row is one-hot.

Combining both directions, we conclude that a row-normalized attention matrix preserves the Frobenius norm of *all* input representations if and only if each of its rows is a one-hot vector. In practical self-attention, the learned attention weights are not row-permutation matrices, so the aggregation step $\mathbf{H}' = \hat{\mathbf{A}}\mathbf{H}$ typically changes the magnitude (Frobenius norm) of the token representations, as claimed in Proposition 1. $\qquad\square$

## B  Baseline Method Pruning Layer Range Tuning

To fairly compare our method against existing pruning approaches, we use perplexity to tune the pruning layer range for all baseline methods. For Mag-sp, Wanda-sp, and LLM-Pruner, the pruning ratio is the same across all pruning layers, whereas FLAP employs an adaptive pruning strategy, allowing different pruning ratios for each layer.

LLaMA3.1-8B[4] and Mistral-7B-v0.3[5] have 32 layers, each with 8 key/value attention heads. Qwen2-7B[6] consists of 28 layers with 4 key/value attention heads per layer, while Gemma2-9B[7] includes 42 layers, each

---

[4]https://huggingface.co/meta-llama/Llama-3.1-8B/blob/main/config.json
[5]https://huggingface.co/mistralai/Mistral-7B-v0.3/blob/main/config.json
[6]https://huggingface.co/Qwen/Qwen2-7B/blob/main/config.json
[7]https://huggingface.co/google/gemma-2-9b/blob/main/config.json

Table 7: The pruning layer range (closed on the left, open on the right) of pruning algorithms for LLaMA3.1-8B, Mistral-7B-v0.3, Qwen2-7B, and Gemma2-9B.

| LLaMA3.1-8B | | |
|---|---|---|
| **Method** | **Starting Layer Index** | **End Layer Index** |
| Mag-sp | 15 | 31 |
| Wanda-sp | 15 | 31 |
| LLM-Pruner | 14 | 30 |
| FLAP | 0 | 32 |
| **Mistral-7B-v0.3** | | |
| **Method** | **Starting Layer Index** | **End Layer Index** |
| Mag-sp | 15 | 31 |
| Wanda-sp | 0 | 32 |
| FLAP | 0 | 32 |
| **Qwen2-7B** | | |
| **Method** | **Starting Layer Index** | **End Layer Index** |
| Mag-sp | 4 | 12 |
| Wanda-sp | 3 | 11 |
| FLAP | 3 | 27 |
| **Gemma2-9B** | | |
| **Method** | **Starting Layer Index** | **End Layer Index** |
| Mag-sp | 2 | 18 |
| Wanda-sp | 30 | 42 |
| FLAP | 3 | 39 |

with 8 key/value attention heads. For LLaMA3.1-8B and Mistral-7B-v0.3, we allocate a KV cache budget of 3/4, while Qwen2-7B and Gemma2-9B have a KV cache budget of 6/7. This requires removing 64 key/value attention heads for LLaMA3.1-8B and Mistral-7B-v0.3, 16 for Qwen2-7B, and 48 for Gemma2-9B.

To determine the number of key/value attention heads to prune in each layer, we restrict the range of removable key/value attention heads to maintain integer pruning layer counts. For LLaMA3.1-8B and Mistral-7B-v0.3, 2 or 4 heads are removed per layer. Qwen2-7B allows 1 or 2 heads per layer, while Gemma2-9B permits 2, 3, 4, or 6 heads per layer. As a result, the pruning layer counts are configured as follows: for LLaMA3.1-8B and Mistral-7B-v0.3, the pruning layer number is 16 or 32; for Qwen2-7B, it is 8 and 16; and for Gemma2-9B, it is 8, 12, 16, or 24. For FLAP, where the pruning ratio can vary across layers, we similarly tune the number of pruned layers to obtain strong perplexity under the same KV cache budget. Specifically, for LLaMA3.1-8B and Mistral-7B-v0.3, the number of pruning layers is set to 12, 16, 20, 24, 28, or 32. For Qwen2-7B, the pruning layer counts are 8, 12, 16, 20, 24, or 28. Similarly, for Gemma2-9B, the options are 9, 12, 18, 24, 30, 36, or 42 layers.

For a fixed-length interval with undefined start and end points, we begin with the starting layer index set to 0. In each step, we increment both the starting layer index and the end layer index by 1 until the end layer index reaches the model's highest layer. The results are presented in Table 7. Note that we also integrate FLAP into HARP to prune FFN parameters. However, for our method we *do not* tune the pruning layer range and instead always prune the highest layers, which makes the comparison with tuned baselines conservative.

## C    Benchmark Details

For our evaluation, we employ the `lm-evaluation-harness` package (version 0.4.7) developed in (Gao et al., 2024). This framework provides a unified interface to commonly used benchmarks and implements task-specific preprocessing and scoring. It is important to note that the `lm-evaluation-harness` provides two accuracy metrics, "acc" and "acc_norm", for the ARC-Challenge, OpenBookQA, and PIQA benchmarks. Unless otherwise stated, for these benchmarks, we report the "acc" accuracy results. Table 8 reports the number of tasks and the number of choices for each discriminative task.

For the GSM8K and TriviaQA benchmarks, this package offers two accuracy metrics: "exact_match,strict-match" and "exact_match,flexible-extract". In our reporting, we use the "exact_match,strict-match" accuracy

Table 8: Dataset Statistics

| Metric | WinoGrande | ARC-Challenge | BoolQ | OpenBookQA | PIQA | MMLU |
|---|---|---|---|---|---|---|
| # Tasks | 1267 | 1172 | 3270 | 500 | 1838 | 11973 |
| # Choices | 2 | 4 | 2 | 4 | 2 | 4 |

results for these benchmarks. The number of tasks for GSM8K, NaturalQuestions, TriviaQA, MATH-hard, and BBH are 1319, 3610, 17944, 1324, and 6511, respectively.

For long-context generation evaluation, we adopt the LongBench benchmark (Bai et al., 2024), which offers a comprehensive suite of tasks designed to assess the ability of language models to handle long sequences across diverse domains. LongBench includes 23 tasks categorized into six groups: (I) Single-Document QA, (II) Multi-Document QA, (III) Long In-Context Learning, (IV) Long Dialogue History Understanding, (V) Code Repository Understanding, and (VI) Long Structured Data Understanding. Each task varies in domain, document length, and complexity, providing a broad challenge spectrum. According to Bai et al. (2024), expert performance ranges from 22% to 89% accuracy, with input lengths spanning from 13k to 167k tokens, and average human solving times between 5 and 13 minutes. This benchmark enables a rigorous evaluation of a model's capacity to reason, retrieve, and generate over extended contexts.

# D    Layer Selection with Heuristic Metric for Pruning Attention Heads

**Hessian-based Metric.**    The Hessian-based metric uses a gradient-based importance score to evaluate the contribution of each self-attention layer to overall model performance. To quantify the importance of a given layer, we define the following:

$$\mathbf{I}(l) = \sum_{s \in \{q,k,v\}} \left( \sum_{i,j} \frac{\partial \mathbf{L}}{\partial \mathbf{W}_{ij}^s} \cdot \mathbf{W}_{ij}^s \right)^2 .$$

Here, $\mathbf{I}(l)$ denotes the importance score of layer $l$, and $s$ refers to the query, key, and value projection matrices in the self-attention mechanism. $\mathbf{W}^s$ represents the weight matrix of projection $s$, while $\frac{\partial \mathbf{L}}{\partial \mathbf{W}_{ij}^s}$ is the gradient of the loss function $\mathbf{L}$ with respect to the weight element $\mathbf{W}_{ij}^s$.

This metric is derived from a first-order Taylor expansion and approximates the effect of removing a layer by computing the squared sum of the element-wise product between weights and their corresponding gradients. Layers with lower importance scores are considered to contribute less to the model's performance.

**Similarity-based Metric.**    The similarity-based metric evaluates the importance of each self-attention layer by measuring the extent to which the layer transforms its input representations. To quantify this transformation, we define the following score:

$$\mathbf{S}_A^{(l)} = 1 - \mathbf{E} \left[ \frac{\langle \mathbf{X}_A^{(l)}, \mathbf{Y}_A^{(l)} \rangle}{\|\mathbf{X}_A^{(l)}\|_2 \|\mathbf{Y}_A^{(l)}\|_2} \right] .$$

Here, $\mathbf{S}_A^{(l)}$ denotes the importance score of layer $l$, $\mathbf{X}_A^{(l)}$ is the input to the self-attention module, and $\mathbf{Y}_A^{(l)}$ is its output, which can be expressed as:

$$\mathbf{Y}_A^{(l)} = \mathbf{X}_A^{(l)} + \text{Attention}\left(\text{LayerNorm}\left(\mathbf{X}_A^{(l)}\right)\right) .$$

This metric captures the degree of change introduced by the self-attention layer via cosine similarity. A higher score indicates a larger transformation between input and output, implying that the layer plays a more critical role in representation learning. Conversely, layers with lower scores perform minimal transformations.

In our experiments, we use these heuristic metrics only for *layer selection* (i.e., to decide which layers are more suitable for pruning) and keep the head-level pruning and rescaling strategy fixed as described in the

main text. This separation of concerns allows us to clearly attribute performance gains to the proposed high-layer attention pruning scheme rather than to a particular choice of heuristic.

