# OpenReview forum: "High-Layer Attention Pruning with Rescaling"
_TMLR — Accepted by TMLR_

### Review · Reviewer_tZup · 2025-11-24

**Summary Of Contributions:**

This paper introduces High-layer Attention Rescaled Pruning (HARP), a structured pruning approach that enhances LLM inference efficiency by selectively removing query and key parameters in upper layers, a strategy grounded in theoretical insights regarding token over-smoothing and reduced discriminative capacity at depth. To mitigate the representational magnitude shifts caused by bypassing attention blocks, the authors propose an adaptive rescaling mechanism using a scalar parameter optimized via a greedy search. Extensive evaluations across multiple architectures demonstrate that HARP outperforms existing baselines like Wanda and FLAP, particularly in generation tasks sensitive to error propagation, while achieving notable inference speedups.

**Audience:**

Yes

**Audience Explanation:**

The strong motivation behind HARP, combined with its algorithmic simplicity and the robust experimental evidence supporting its efficacy, makes this work highly relevant to the TMLR audience. It will be of particular interest to researchers focused on efficient LLM inference and model compression. Moreover, the theoretical insights and empirical findings presented in this study provide a solid foundation that will likely facilitate and inspire subsequent research in the field.

**Broader Impact Concerns:**

I do not see any broader impact concerns that would require a specific ethical statement.

**Claims And Evidence:**

Yes

**Claims Explanation:**

The claims presented in this paper are supported by robust and comprehensive evidence. The authors establish a convincing motivation for high-layer attention pruning by effectively synthesizing theoretical analysis, specifically regarding the reduced discriminative capacity of high-layer heads due to over-smoothing and the computational bottlenecks of long-context modeling, with existing literature. The proposed HARP framework is rigorously designed to align with this motivation and introduces a clear solution (adaptive rescaling) to address the specific issue of representational magnitude shifts caused by pruning. The empirical validation is extensive, demonstrating the method's consistent superiority over baselines across a diverse set of LLM architectures (e.g., LLaMA3.1, Mistral, Qwen) and 27 datasets. Furthermore, the inclusion of detailed ablation studies regarding layer selection strategies and efficiency analysis provides a clear confirmation of the algorithm's effectiveness and practical viability.

**Requested Changes:**

While I positively evaluate the strong motivation, the simplicity and effectiveness of the proposed algorithm, and the solid empirical evidence presented in this work, I believe the following adjustments are necessary to meet the acceptance criteria.

**Critical to securing a recommendation for acceptance:**

1.  Cost comparison with baselines: In Section 3.5 (Page 9), the authors mention that the $\alpha$ search for HARP requires approximately two hours on an NVIDIA H100 GPU. To rigorously assess the efficiency of the proposed method, it is essential to compare this computational cost against the setup or calibration times required for the baseline pruning algorithms (Mag-sp, LLM-Pruner, Wanda-sp, and FLAP). Providing a cost comparison will allow for a clearer evaluation of the trade-off between HARP’s performance gains and its additional search overhead.

**Would strengthen the work:**

2.  Table captions: Please explicitly state the meaning of the bolded text (e.g., "Bold indicates the best performance") in the captions of all performance tables (Tables 1, 2, 3, etc.) to ensure the results are immediately interpretable.

3.  Clarification of Figure 3: In Figure 3, the green dashed line appears to represent the relative speed improvement between the original and pruned models. Please expand the figure caption to explicitly describe what this line and the secondary y-axis represent to aid reader comprehension.

4.  Expansion of conclusion: The current Conclusion section is relatively brief. I suggest expanding it to go beyond a simple summary of the algorithm and results. It would be beneficial to discuss the expected impact of the proposed method on real-world tasks, explicitly address any known limitations, and outline potential directions for future work.

---

> ### Author Response · Authors · 2025-11-29
>
> 1. Cost comparison with baselines: In Section 3.5 (Page 9), the authors mention that the $\alpha$ search for HARP requires approximately two hours on an NVIDIA H100 GPU. To rigorously assess the efficiency of the proposed method, it is essential to compare this computational cost against the setup or calibration times required for the baseline pruning algorithms (Mag-sp, LLM-Pruner, Wanda-sp, and FLAP). Providing a cost comparison will allow for a clearer evaluation of the trade-off between HARP’s performance gains and its additional search overhead.
>
> **Rebuttal:**
>
> In HARP, we perform a layer-wise search, requiring 10 inferences per layer. For an 8-layer model, this amounts to 80 inferences in total, whereas the baseline needs only a single inference. We acknowledge that this is more computationally expensive than the baseline; however, each inference takes less than one minute, and this search is performed only once to determine a good α. Compared with training-based methods, our approach still requires substantially less computation, so we believe this overhead is reasonable.
>
> 2. Table captions: Please explicitly state the meaning of the bolded text (e.g., "Bold indicates the best performance") in the captions of all performance tables (Tables 1, 2, 3, etc.) to ensure the results are immediately interpretable.
>
> **Rebuttal:**
>
> Thanks for the suggestion. We have included it in the updated version. Please check it.
>
> 3. Clarification of Figure 3: In Figure 3, the green dashed line appears to represent the relative speed improvement between the original and pruned models. Please expand the figure caption to explicitly describe what this line and the secondary y-axis represent to aid reader comprehension.
>
> **Rebuttal:**
>
> Thanks for the suggestion. We have added an additional description in the updated version. Please check it.
>
> 4. Expansion of conclusion: The current Conclusion section is relatively brief. I suggest expanding it to go beyond a simple summary of the algorithm and results. It would be beneficial to discuss the expected impact of the proposed method on real-world tasks, explicitly address any known limitations, and outline potential directions for future work.
>
> **Rebuttal:**
>
> Thanks for the suggestion. We have expanded the conclusion in the updated version. Please check it.

---

### Review · Reviewer_egY7 · 2025-11-27

**Summary Of Contributions:**

This work introduces High-layer Attention Rescaled Pruning (HARP), a novel training-free structured pruning algorithm for Large Language Models (LLMs). HARP strategically removes attention heads in higher model layers, arguing they have reduced discriminative capacity due to increasing token similarity. A key feature is an adaptive rescaling parameter $\alpha$ introduced into the attention residual block to maintain token representation magnitude post-pruning and mitigate performance loss. Experiments across 27 tasks on different LLMs consistently show HARP outperforms existing structured pruning methods, offering significant speed improvements, especially for long-context tasks.

Strength:
- The core idea of strategically pruning attention heads in the higher layers is a novel and effective design choice for structured pruning. It challenges conventional approaches that prune heads indiscriminately or based solely on per-layer importance metrics.
- This work demonstrates superior experimental results. The proposed HARP significantly outperforms all compared baselines in generation tasks across all four tested LLMs (LLaMA3.1-8B, Mistral-7B-v0.3, Qwen2-7B, and Gemma2-9B).

Weakness:
The determination of the optimal alpha is achieved via a greedy search strategy (grid search, layer-by-layer) evaluating perplexity, which is a heuristic and sub-optimal approach for finding a globally optimal configuration.
The notations in the method part are not clear enough. For example, in equation (3), I cannot directly understand what do W_v and W_o refer to. Similar situations exist in other formulas.

**Audience:**

Yes

**Audience Explanation:**

The researchers that focus on efficient LLM reasoning will be interested on the finding of this paper.

**Broader Impact Concerns:**

There are not concerns that that would require adding a Broader Impact Statement.

**Claims And Evidence:**

Yes

**Claims Explanation:**

Generally, the claims and contributions in the paper is supported by accurate, convincing and clear evidence.

**Requested Changes:**

A table to compare with HARP with other related pruning methods to make related work comparison clearer.
Compare the proposed greedy search strategy with other possible hyperparameter search strategies.
Experiment on different LLM size. Currently, the parameter scale of tested LLMs is between 7B and 9B. It is necessary to conduct experiments on both smaller and larger LLMs to verify the robustness of the proposed pruning strategy.

---

> ### Author Response · Authors · 2025-11-29
>
> We thank the reviewer for the helpful suggestions and feedback. We address the concerns below:
>
> 1. The determination of the optimal alpha is achieved via a greedy search strategy (grid search, layer-by-layer) evaluating perplexity, which is a heuristic and sub-optimal approach for finding a globally optimal configuration. The notations in the method part are not clear enough. For example, in equation (3), I cannot directly understand what do W_v and W_o refer to. Similar situations exist in other formulas.
>
> **Rebuttal:**
>
> We're sorry for the confusion. In a standard attention mechanism, three projection matrices $W_q, W_k$, and $W_v$ are used to compute the queries, keys, and values, respectively, and a fourth matrix $W_o$ is applied afterward to project the attention output back to the model dimension.
>
> 2. A table to compare with HARP with other related pruning methods to make related work comparison clearer.
>
> **Rebuttal:**
>
> Thanks for the suggestion. We have included the table in our updated version. Please check it.
>
> 3. Compare the proposed greedy search strategy with other possible hyperparameter search strategies.
>
> We introduce a search strategy to determine a single fixed α shared across all layers. The results are shown below.
>
> | Method                         | WinoGrande | ARC-Challenge | BoolQ | OpenBookQA |  PIQA |  MMLU |   Avg |
> |--------------------------------|-----------:|--------------:|------:|-----------:|------:|------:|------:|
> | fixed-alpha for all layers     |     72.30  |         45.73 | 81.99 |      27.80 | 73.56 | 58.27 | 59.94 |
> | adaptive alpha for each layer  |     **76.64**  |     **51.02** | **79.82** |      **31.80** | **78.89** | **61.96** | **63.35** |
>
>
> 4. Experiment on different LLM size. Currently, the parameter scale of tested LLMs is between 7B and 9B. It is necessary to conduct experiments on both smaller and larger LLMs to verify the robustness of the proposed pruning strategy.
>
> Thanks for the suggestion. We found that for small models such as Qwen/Qwen2-1.5B, the GQA group size is too small (2), and the baseline implementation does not support our current 25% configuration. For larger models, the baseline implementation does not yet support distributed or quantized inference, so a single GPU does not have enough memory to run them.

---

### Review · Reviewer_KJ8C · 2025-11-28

**Summary Of Contributions:**

This manuscript proposes a training-free structured prunnig algorithm called HARP (High-layger Attention Rescaled Pruning). It removes the attention heads, specifically the query and key parameters in the higher layers. After pruning, rescaling in the residual block is used to rescale the magnitude. The effectiveness is evaluated with LLaMA3.1-8B, Mistral-7B-v0.3, Qwen2-7B, and Gemma2-9B on 27 downstream tasks.

**Audience:**

Yes

**Audience Explanation:**

The topic of LLM pruning is of great empirical interest in today's large model age.

**Broader Impact Concerns:**

N/A.

**Claims And Evidence:**

Yes

**Claims Explanation:**

The algorithm is evaluated with four LLMs on many downstream tasks with popular training-free methods.

**Requested Changes:**

1. The writing is not always clear. For example, the sentence "It is important to note that the official implementation does not support GQA-based LLMs. Therefore, we use a modified implementation for our experiments" even does not tell the readers it is about FLAP. Please go through a more careful proofreading.

2. The reported performance is promising. However, the authors did not provide any code to reproduce the results. I will not suggest acceptance of a paper which claims SOTA but provides no code.

3. The first proof environment in Page 20 is unprofessionally written. Proof of what?

---

> ### Author Response · Authors · 2025-11-28
>
> We thank the reviewer for the helpful suggestions and feedback. We address the concerns below:
>
> 1. The writing is not always clear. For example, the sentence "It is important to note that the official implementation does not support GQA-based LLMs. Therefore, we use a modified implementation for our experiments" even does not tell the readers it is about FLAP. Please go through a more careful proofreading.
>
> **Rebuttal:**
>
> We thank the reviewer for the helpful suggestion. We have revised this sentence to make it clear in the updated version. Please check it!
>
> 2. The reported performance is promising. However, the authors did not provide any code to reproduce the results. I will not suggest acceptance of a paper which claims SOTA but provides no code.
>
> **Rebuttal:**
>
> We have provided the code in the updated submission. Please check it!
>
> 3. The first proof environment in Page 20 is unprofessionally written. Proof of what?
>
> **Rebuttal:**
>
> We show that higher-layer attention heads are less important by analyzing how token similarity patterns evolve across different layers in page 20.

---

> > ### Comment · Reviewer_KJ8C · 2025-11-29
> >
> > Regarding Point 1: I suggest reviewing the manuscript more thoroughly to enhance overall clarity, rather than only addressing this particular issue. For example, I came across the following sentence at random: "Clearly, all Y within the M space are prone to over-smoothing, as the representations of all tokens in Y are the same."Could you clarify what you mean by "the same"? This term is too vague. Please elaborate with more concrete and specific wording. In addition, why are the revisions not highlighted or explicitly marked? Making the changes stand out would help reviewers identify them more easily.
> >
> > Regarding Point 2:​ The submitted code does not include a README file, and inline comments are in Chinese. This makes it difficult for non-Chinese-speaking readers to understand and use the code. I strongly encourage the authors to treat the supplementary materials with the same level of care as the main manuscript, since they play a crucial role in ensuring reproducibility and demonstrating the rigor of the research.
> >
> > Regarding Point 3: Do you think you should at least mention "This is the proof of Proposition X"?
> >
> > I am not satisfied with the revison. The reason is simple: In the era of large language models, the time needed to properly refine a manuscript has been considerably shortened, thanks to AI-assisted writing and editing tools. This means that producing a well-polished paper should require only a modest additional effort from the authors. Yet, despite these readily available advantages, the current submission falls short in terms of clarity, completeness, and overall care. It is hard to see why one would not take the opportunity to treat their own work with greater seriousness and diligence, especially when doing so has become so much more feasible.
> >
> > I suggest the authors review the manuscript more thoroughly to enhance overall clarity.

---

> > > ### Author Response · Authors · 2025-11-29
> > >
> > > For point 2. It's about the dataset, it's not about the comments...
> > >
> > > You can use bash ex.sh to run the experiments.
> > >
> > >
> > > For points 1 and 3, we have polished the paper based on your suggestions.

---

> ### Comment · Reviewer_KJ8C · 2025-11-30
>
> > *For point 2. It's about the dataset, it's not about the comments...*
>
> **Re:** In your code (`main.py`):
>
> ```
> bytes_per_mb = 1024 * 1024  # 每MB的字节数
> tensor_size_mb = 0       # 期望的Tensor大小，以MB为单位
> element_size_bytes = 4      # float32元素的大小，以字节为单位
> ```
>
> My intention was simply to give you time to properly revise the paper and code. I wasn’t opposed to accepting it, but the manuscript is quite rough at least to me.
>
> Several aspects are not professional or reader-friendly.
> - The code lacks comments and a README.
> - Proposition 1 is stated without an explicit proof. Besides, it is more like a simple fact rather than a proposition.
> - Proposition 2 is also without proof. You said `A detailed proof of Proposition 2 is also given in Appendix A (“Proof of Proposition 2”)`, but **where**?
>
> What frustrated me was the author’s attitude. These improvements should not require much time. If the author keeps being this careless, I won’t continue playing this game with you.

---

> > ### Author Response · Authors · 2025-11-30
> >
> > We are sorry we used an out-dated version.
> >
> > We added the readme and deleted the Chinese comments.
> >
> > We rewrite Proposition 1.

---

> ### Comment · Reviewer_KJ8C · 2025-11-30
>
> Thanks for your prompt revision.
>
> In the original manuscript, you did not provide a proof of Proposition 2 (now renamed as Proposition 1 in the revision), so it may take some time to verify its correctness.
>
> In addition, there is no need to write “This is the proof of Proposition 1” inside the LaTeX proof environment. I mentioned this in earlier comments only because I couldn’t find any proofs for Proposition 1 or Proposition 2, and I assumed your proof might correspond to one of them. To be clear, you can use
> `\begin{proof}[Proof of Proposition 1] ... \end{proof}`
> to specify which statement you are proving.
>
> Moreover, your content for proof of Proposition 2 is entirely new, so it should all be marked in blue. I don’t want to say more about this clver trick.

---

> > ### Author Response · Authors · 2025-11-30
> >
> > Done as you suggested. Thanks!

---

### Decision · Action_Editor_xsm7 · 2026-01-15

**Recommendation:** Accept with minor revision

**Additional Comments:**

I highly recommend the authors to revise the paper carefully by taking all the recommendations from the reviewers. For example, please proof read the presentation, and provide cost comparison with baselines (the authors have discuss this in the rebuttal but do not seem to incorporate it into the revision).

**Audience:**

Yes

**Audience Explanation:**

Pruning is an important and interesting method to scaling up LLM training and inference. The findings in the paper would be found valuable by LLM researchers.

**Claims And Evidence:**

Yes

**Claims Explanation:**

The claims are generally supported by accurate and convincing evidence, mainly due to the following:
1. The proposed method is evaluated with multiple LLMs on many downstream tasks, and compared with related baselines.
2. The motivation is convincing. The proposed method is rigorously designed and clearly presented.

---

> ### Author Response · Authors · 2026-01-15
>
> Hi AE,
>
> We have now added a cost comparison with the baselines in Section 3.1. After proofreading and revising the manuscript accordingly, we have submitted an updated version. Thank you for recognizing our work and for the time and effort you invested in reviewing the paper—we sincerely appreciate it.
>
> Best,
>
>  The authors